# Antibody Titer Kinetics and SARS-CoV-2 Infections Six Months after Administration with the BNT162b2 Vaccine

**DOI:** 10.3390/vaccines9111357

**Published:** 2021-11-19

**Authors:** Davide Ferrari, Nicola Clementi, Elena Criscuolo, Alessandro Ambrosi, Francesca Corea, Chiara Di Resta, Rossella Tomaiuolo, Nicasio Mancini, Massimo Locatelli, Mario Plebani, Giuseppe Banfi

**Affiliations:** 1Scienze Chimiche della Vita e della Sostenibilità Ambientale (CVSA) Department, University of Parma, 43121 Parma, Italy; 2Laboratory of Microbiology and Virology, Vita-Salute San Raffaele University, 20158 Milan, Italy; clementi.nicola@hsr.it (N.C.); criscuolo.elena@hsr.it (E.C.); mancini.nicasio@hsr.it (N.M.); 3IRCCS Ospedale San Raffaele, Laboratory Medicine Service, 20158 Milan, Italy; corea.francesca@hsr.it (F.C.); diresta.chiara@hsr.it (C.D.R.); locatelli.massimo@hsr.it (M.L.); 4Surgery and Medicne Depertment, University Vita-Salute San Raffaele, 20158 Milan, Italy; ambrosi.alessandro@hsr.it (A.A.); tomaiuolo.rossella@hsr.it (R.T.); banfi.giuseppe@hsr.it (G.B.); 5Department of Laboratory Medicine, Padua University School of Medicine, 35100 Padua, Italy; mario.plebani@unipd.it; 6Laboratory of Clinical Chemistry and Microbiology, IRCCS Orthopedic Institute Galeazzi, 20161 Milan, Italy

**Keywords:** COVID-19, serological test, mRNA vaccine, Roche Anti-SARS-CoV-2-S, immune response, vaccination

## Abstract

Background: Studies reporting the long-term humoral response after receiving the BNT162b2 COVID-19 vaccine are important to drive future vaccination strategies. Yet, available literature is scarce. Covidiagnostix is a multicenter study designed to assess the antibody response in >1000 healthcare professionals (HCPs) who received the BNT162b2 vaccine. Methods: Serum was tested at time-0 (T_0_), before the first dose, T_1_, T_2,_ and T_3_, respectively, 21, 42, and 180 days after T_0_. Antibodies against the SARS-CoV-2 nucleocapsid-protein were measured to assess SARS-CoV-2 infections, whereas antibodies against the receptor-binding domain of the spike protein were measured to assess the vaccine response. Neutralization activity against the D614G, B.1.1.7, and B.1.351 variants were also analyzed. Results: Six months post-vaccination HCPs showed an antibody titer decrease of approximately 70%, yet, the titer was still one order of magnitude higher than that of seropositive individuals before vaccination. We identified 12 post-vaccination infected HCPs. None showed severe symptoms. Interestingly, most of them showed titers at T_2_ above the neutralization thresholds obtained from the neutralization activity experiments. Conclusion: Vaccination induces a humoral response which is well detectable even six months post-vaccination. Vaccination prevents severe COVID-19 cases, yet post-vaccination infection is possible even in the presence of a high anti-S serum antibody titer.

## 1. Introduction

As of 6 October 2021, the COVID-19 pandemic, caused by Severe Acute Respiratory Syndrome Coronavirus-2 (SARS-CoV-2), is responsible for more than 230 million infected people and nearly five million deaths [1]. The BNT162b2 mRNA COVID-19 vaccine (Comirnaty^®^, Pfizer BioNTech) was one of the first vaccine formulations to be developed and approved under an Emergency Use Authorization (EUA) on 11 December 2020 by the FDA, and on 21 December by the European Union; on 23 August it received full approval by the FDA [2]. Its administration protocol has been based on a double dose approach which proved to be effective in preventing 95% of COVID-19 cases [3,4]. However, Pfizer’s CEO recently announced the need of a third vaccine dose, likely within 12 months of the first dose [5], which has been recently authorized in several countries for some frail population [6]. The Comirnaty vaccine exploits mRNA, encoding the SARS-CoV-2 full-length spike protein (S-protein). In order to promote the production of receptor binding domain (RBD) neutralizing antibodies, the mRNA nucleotide sequence has been modified by two proline mutations to lock the S-protein in the prefusion conformation [7,8]. Anti-S antibodies can be detected with quantitative serological tests, but also with neutralization tests, which measure the ability of the subject’s antibodies to protect cells from infection [9,10,11,12,13]. Several studies have monitored the antibody response after the first dose of the Comirnaty vaccine observing a higher anti-S antibody response in persons previously infected by SARS-CoV-2 compared to subjects never infected by SARS-CoV-2 [9,10,11,12,14,15]. A few studies also evaluated the response after the second vaccine administration, however, the follow-up of enrolled subjects was up to 3 months only [16,17,18] and the number of participants was limited (≤400). To the best of our knowledge only one study followed the serological response after a 6 month follow-up period yet the cohort was limited (*n* = 33) and was injected with a different vaccine (Moderna) [19]. A longer follow-up period and large cohorts of subjects are thus sought to better assess the antibody kinetics and the neutralizing activity of sera against the emerging virus variants of concern, as well as to better inquire into the need of a third vaccine dose.

We took advantage of our ongoing multicenter longitudinal study (Covidiagnostix), funded by the Italian Ministry of Health, to investigate the antibody responses in a six month period of over 1000 healthcare professionals (HCPs) injected with the Comirnaty vaccine [20]. The objective of the study was the evaluation of the antibody response induced by the current Comirnaty vaccine administration protocol in different sex/age groups, as well as in subjects seropositive/seronegative for anti-SARS-CoV-2 antibodies before vaccine administration. In addition, as serum antibody neutralizing activity represents, so far, the best surrogate of protection for COVID-19, we evaluated the sera’s neutralizing activity stratified according to their anti-RBD-S-protein antibody titer, at different time points. We speculate that inferring anti-RBD-S-protein antibody titer thresholds, possibly related to serum neutralizing activity, could bring new important information for the implementation of new vaccine administration protocols [5,6].

## 2. Materials and Methods

### 2.1. Covidiagnostix Study

The Covidiagnostix is an ongoing multicenter study, approved by the San Raffaele Hospital, Milan, Italy, Institutional Ethical Review Boards (CE:199/INT/2020), which aims to monitor the antibody response of a population of HCPs who were offered the BNT162b2 mRNA COVID-19 (Comirnaty) vaccine. 

### 2.2. Inclusion Criteria and Methodology

This study included 1052 HCPs from the San Raffaele Hospital, Milan, Italy. All of the HCPs received two doses of the BNT162b2 vaccine (21 ± 1 day’s interval between the two doses) during January and February 2021; no exclusion criteria were applied. Blood samples were withdrawn for serological evaluation, as previously described [21] at: time 0 (T_0_), 1–2 min before receiving the first vaccination dose. Time 1 (T_1_), 21 ± 2 days after T0, before (1–2 min) the injection of the second dose. Time 2 (T_2_), 42 ± 4 days after T_0_. Time 3 (T_3_), 180 ± 10 days after T0. At T_0_, samples were tested for the presence of SARS-CoV-2 antibodies using the Roche Anti-SARS-CoV-2, an electrochemiluminescence immunoassay (ECLIA) (Sensitivity: 100%; Specificity: 99.8%, by adopting the manufacturer’s suggested cutoff of 1 U/mL), on a COBAS 601 platform (Roche, Basel, Switzerland), targeted on total Immunoglobulins (IgTot: IgA, IgG and IgM) against the viral nucleocapsid protein (N-protein) [22,23]. Tests results between 0.165 and 1 U/mL were considered dubious [24], thus, when available, previous diagnostic tests (RT-PCR tests) were used to discriminate between SARS-CoV-2 previously infected and non-previously infected individuals. Thanks to an instrument query, upon dubious/positive results (>0.165 U/mL), samples were further tested on the same platform with the Roche anti-SARS-CoV-2-S test (Roche, Basel, Switzerland): an electrochemiluminescence immunoassay (ECLIA) detecting IgTot against the receptor binding domain (RBD) of the viral S-protein. The quantification range is between 0.4 and 250.0 U/mL, which is further extended to 2500.0 U/mL by a 1:10 dilution of the sample automatically performed by the instrument. Specificity and sensitivity (≥14 days after diagnosis) are 99.98% and 98.8%, respectively, by adopting the manufacturer’s suggested cutoff of 0.8 U/mL. At T_1_, T_2_, and T_3_ the samples were tested for the presence of anti-Spike protein antibodies with the Roche anti-SARS-CoV-2-S test. At T_3_, when needed, samples were also tested for the presence of antibodies against the N-protein (Roche Anti-SARS-CoV-2). It must be noted that for the anti-RBD-S titers the manufacturer claims a Unit per milliliter (U/mL) to Binding Antibody Units per milliliter (BAU/mL, proposed by the WHO to standardize any device to the WHO International Standard) conversion factor equal to 1. Therefore, anti-RBD-S titers expressed in U/mL throughout the paper correspond to BAU/mL [25].

### 2.3. COVID-19 Diagnostic Data

From the beginning of the COVID-19 pandemic, as part of a follow-up institutional program, real-time PCR swab tests were performed both routinely and whenever a HCP showed symptoms consistent with COVID-19. Nasopharyngeal swabs were analyzed using the Tib-Molbiol 2019-nCoV real-time reverse-transcription PCR Kit (cat# 61011896) on a Roche Cobas Z480 thermocycler (Roche Diagnostic, Basel, Switzerland). RNA purification was performed using the Roche Magna pure system (cat# A42352) [26]. 

### 2.4. Viruses and Cells

Vero E6 (Vero C1008, clone E6-CRL-1586; ATCC) cells were cultured in Dulbecco’s Modified Eagle Medium (DMEM) supplemented with non-essential amino acids (NEAA), penicillin/streptomycin (P/S), HEPES buffer, and 10% (*v/v*) fetal bovine serum (FBS). Three clinical isolates of SARS-CoV-2 were obtained and propagated in Vero E6 cells as previously described [27]: D614G (hCoV-19/Italy/UniSR1/2020; GISAID Accession ID: EPI_ISL_413489), B.1.1.7 (Alpha) (19/Italy/LOM-UniSR7/2021; GSAID Accession ID: EPI_ISL_1924880), B.1.351 (Beta) (hCoV-19/Italy/LOM-UniSR6/2021, GISAID Accession ID: EPI_ISL_1599180).

### 2.5. Micro-Neutralization Experiments

Samples from 48 different HCPs underwent a neutralization activity test at T_2_ and T_3_. The 48 HCPs’ samples were randomly chosen, at T_1_, to cover a broad antibody titer range (Roche Anti-SARS-CoV-2-S) from below the detection level (<0.4 U/mL) to above the high instrument limit (>2500 U/mL). Samples showing antibody titers exceeding the upper instrument limits were further diluted with a pool of pre-pandemic sera to obtain test results within the quantification range.

Vero E6 cells (4 × 105 cells/mL) were seeded into 96-well plates 24 h before the experiment was performed at 95% cell confluency for each well. Serum samples were decomplemented by incubation at 56 °C for 30 min, diluted to 1:80 and 1:160 and incubated with SARS-CoV-2 strains at 0.01 multiplicity of infection (MOI) at 1:80 and 1:160 dilution for 1 h at 37 °C [28]. Virus-serum mixtures, as well as positive infection control, were applied to Vero E6 monolayers after washing cells with PBS 1X, and virus adsorption was carried out at 37 °C for 1 h. Cells were then washed with PBS 1X to remove cell-free virus particles and virus-containing mixtures, while controls were replaced with complete DMEM supplemented with 2% FBS. The plates were incubated at 37 °C in the presence of CO2 for 72 h. The experiments were performed in triplicate. Neutralization activity was evaluated by comparing the percentage of cytopathic effect (CPE) detected in the virus-serum mixtures with the positive infection control. Neutralization activity was ranked as follow: 100%, 66.7%, 33.3%, and 0% if all, two, one, and none of the triplicate experiments showed neutralization, respectively.

Live images were acquired (Olympus CKX41 inverted phase-contrast microscopy), CPE was assessed using a scoring system (0 = uninfected; 0.5 to 2.5 = increasing number/area of plaques; 3 = all cells infected) to evaluate all of the tested conditions. Infection control (score 3) was set as 0% infection inhibition and uninfected cells (score 0) as 100% infection inhibition. The whole surface of the wells was considered for the analysis (5x magnification). All conditions were tested in triplicate.

### 2.6. Statistical Analysis

Observed categorical measures were summarized by means of frequencies or percentages. Numeric observed measures were summarized with mean and standard deviations (SD), or median and Inter Quartile Range (IQR) when in the presence of an excess of censored data. 

Differences in antibody titer between genders were assessed by the Mann–Whitney U statistic test. The relationship between antibody titer and age classes was analyzed by means of the exact Jonckheere trend test [29]. To investigate the antibody titer variation between T_2_ and T_3_ as function of the antibody titer at T_2_, a robust linear model based on MM estimates with high breakdown point was fitted to keep into account possible outlier values [30]. The kinetic of the mean antibody titer over time was fitted by Generalized Additive Mixed Effect Models with spline functions as smooth term and random effect for subject.

For each of the three variants, Receiver Operating Characteristic (ROC) curves and the associated Area Under the Curve (AUC) for the neutralizing activity was computed assuming a linear mixed model with fixed effect for test and crossed random effects for subject and time. More details about this approach can be found in [30]. For graphical purposes, ROC curves were fitted by Kernel density estimates. One sample was defined as “neutralizing” if at least one of the triplicates showed neutralizing activity. Best threshold was defining according to the “top-left” rule, that is the threshold minimizing (1 − sensitivity)^2^ + (1 − specificity)^2^.

Exact *p*-values were computed by means of permutation methods to avoid any distributional approximation. All analyses were performed in R environment (ver. 4.1.1).

## 3. Results

### 3.1. Serological Evaluation at T_0_

Of the 1052 HCPs tested, 81 showed the presence of anti-N antibody (>1 U/mL), consistent with previous SARS-CoV-2 infection, whereas 18 showed dubious results (0.165–1 U/mL). Diagnostic information collected as part of the institutional follow-up program showed that only 1 HCP of the above mentioned 18 had previously experienced COVID-19, bringing the number of seropositive subjects to 82 (7.8%). Seropositive HCPs were further tested for the presence of anti-S antibodies. Their anti-RBD-S antibody titers median value was 58.3 (IQR: 135.9) U/mL with females showing slightly higher antibody titers (Table 1).

### 3.2. Serological Evaluation at T_1_

#### 3.2.1. Seropositive Group

As observed in previous studies [11,12,14,16,31] the 82 HCPs identified at T0 as seropositive showed an exceptional antibody response that was above the 2500 U/mL instrument limit in 71 (86.6%) subjects (Table 1). The remaining 11 HCPs showed a median of antibody titers equal to 658 U/mL (IQR 1433 U/mL).

#### 3.2.2. Seronegative Group

Of the 970 HCPs seronegative at T0, 12 were retrospectively excluded from this group because they became infected by SARS-CoV-2 after or during the vaccination protocol and will be discussed in Section 3.5. The remaining 958 HCPs, except 34 (3.5%), all showed a post-vaccination humoral response at T_1_. Their anti-RBD-S antibody titers were of the same order of magnitude as those observed in the seropositive group at T0 (Table 1) and decreased significantly with age (*p* < 0.00001) (Figure 1). In contrast, no significant differences were observed when comparing males and females. Only one subject showed an antibody titer >2500 U/mL. The 17 HCPs showing dubious anti-N antibodies results, listed as seronegative in Section 3.1, did not show the exceptionally high response (>2500 U/mL) observed in seropositive subjects.

### 3.3. Serological Evaluation at T_2_

#### 3.3.1. Seropositive Group

Compared to T_1_ the number of seropositive HCPs showing antibody titers >2500 U/mL increased from 71 (86.6%) to 78 (95.1%) (Table 1). The remaining four HCPs showed antibody titers at T_2_ higher than their corresponding T_1_ titers (median value 1045 U/mL, IQR 1247.3 U/mL).

#### 3.3.2. Seronegative Group

The antibody titers of the seronegative group, measured 21 days after receiving the second vaccine dose, increased approximately two-orders of magnitude when compared to T_1_ (Table 1, Figure 1). The number of HCPs that did not even respond to the second dose was three (0.3%). In contrast, 283 (29.5%) HCPs showed antibody titers >2500 U/mL. As observed for T_1_, the antibody titers decreased significantly with age for both males (*p* = 0.00857) and females (*p* < 0.00001) (Figure 1), whereas no significant differences were observed when comparing the two genders.

### 3.4. Serological Evaluation at T_3_

#### 3.4.1. Seropositive Group

Six months after the first vaccination dose the median values for both males and females were still above the 2500 U/mL instrument limit (Table 1). However, the number of HCPs showing antibody titers within the assay reading range increased from 4 (4.9%) at T_2_ to 36 (43.9%) at T_3_ (median value 1262.5 U/mL, IQR 977.5 U/mL), 10 of which were males and 26 females (Table 1). 

#### 3.4.2. Seronegative Group

At T_3_ the number of HCPs showing antibody titers >2500 U/mL dropped from 283 (29.5%) at T_2_ to 17 (1.8%), showing a general remarkable decrease in antibody titers with time (Figure 1). The median values for the different age groups showed a large anti-RBD-S titer drop at T_3_ when compared with T_2_ (Table 1). The antibody titers measured at T_3_ decreased significantly with age for both males (*p* < 0.00001) and females (*p* < 0.00001) (Figure 1), whereas no significant differences were observed when comparing the two genders. Out of the three HCPs that didn’t even respond to the second vaccination dose (T_2_), two of them showed detectable, albeit low, antibody titers at T_3_.

### 3.5. Post-Vaccination Infections

In contrast to the general downward antibody titers’ trend, 67 (6.4%) HCPs (42 females and 25 males; 52.6 ± 11.2 years, all seronegative at T0) showed increased antibody titers at T_3_. Among them, seven individuals (four females and three males) showed antibody titers at T_3_ above the 2500 U/mL instrument limit, whereas the remaining 60 showed a median T_2_ to T_3_ titer increase equal to 167 U/mL (IQR 375.5 U/mL). The T_3_ sera of the 67 HCPs were tested for the presence of anti-N antibody: five were above the 1 U/mL cutoff level (Appendix A: subjects 1, 4, 5, 9, 10), thus consistent with post-vaccination infection, whereas two showed dubious results (0.165–1 U/mL) (Appendix A: subjects 3, 8). Post-vaccination infection was confirmed for the latter two HCPs by the presence of positive RT-PCR swab tests performed in the post-vaccination period. Positive swab tests were also present for three of the five HCPs showing anti-N antibody titer >1 U/mL.

Out of the 1052 HCPs involved in the study, 65 (6.2%) HCPs (41 females and 24 males, 45.3 ± 13.3 years old) showed anti-RBD titers above >2500 U/mL, both at T_2_ and T_3_, and were listed as “indefinite”. Forty-six of them were seropositive at T0, the remaining 19 subjects, seronegative at T0, were tested for the presence of anti-N antibodies at T_3_. Out of 19 HCPs, two were above the 1 U/mL cutoff limit (Appendix A: subjects 6, 7) (their post-vaccination infection was also confirmed by previous diagnostic tests), whereas two subjects had dubious anti-N antibody titers. One of the latter had a previous positive swab tests in the post-vaccination period (Appendix A: subject 2), whereas the second one had no history of COVID-19, showed dubious anti-N antibody titers also at T0, yet did not exhibit the distinguished high antibody titer response at T_1_ observed in seropositive subjects [11,12,14,16,32].

Two more HCPs, (seronegative at T0) were infected after receiving the second dose (Appendix A: subjects 11, 12). They did not show T_3_ minus T_2_ Ab titers <0 and were identified thanks to post-vaccination nasopharyngeal swab-RT-PCR test results available as part of the institutional follow-up program. Both females were infected shortly after receiving the second dose (7 and 14 days, respectively). As expected, their T_3_ serum showed the presence of anti-N antibodies. As a control group, 300 HCPs (seronegative at T0) showing a T_2_ to T_3_ decreased antibody titer were also tested for the presence of anti-N antibody. All of them showed both negative results and no presence of post-vaccination positive swab tests.

Altogether, 12 (1.2%) HCPs were infected after receiving the Comirnaty vaccine. According to the available diagnostic information, one subject was infected between the first and the second dose, two were infected between 7 and 14 days after the second dose, seven were infected between 57 and 100 days after the second dose, whereas two HCPs were oblivious of having been infected. Half of the subjects were asymptomatic, whereas the other six claimed mild anosmia and ageusia, accompanied in two cases by cold and in one case by a generalized pain. Eight post-vaccination infected HCPs reported the presence of a SARS-CoV-2 positive family member (not vaccinated) at the time of their infection (Appendix A). Interestingly, 10 out of 12 subjects had antibody titers at T_2_ above 1000 U/mL, and seven of them were above 2000 U/mL (Appendix A). Only two had a rather low titer (<400 U/mL). The 12 HCPs performed, within the hospital, different tasks and belonged to different medical departments (Appendix A).

### 3.6. Antibody Titer Decrease between T_2_ and T_3_

To quantify the antibody titer decrease occurring after the anti-RBD-S antibody titer peaks which, according to Favresse et al. [16] is represented by T_2_, we plotted the T_2_ values of the 958 seronegative groups’, versus their subsequent antibody titer variations (T_3_-T_2_) (Figure 2). We obtained a statistically significant (*p* < 0.0001) linear decrease with a slope equal to −0.62 (CI95%: −0.59 to −0.66). However, the value was biased by the several antibody titers above the 2500 U/mL instrument limit (*n* = 213). Excluding these values from the linear regression analysis we obtained a slope equal to −0.73 (CI95%: −0.69 to −0.77). By further removing the small subset of HCPs (*n* = 62) which, in contrast to the general trend, increased their antibody titer from T_2_ to T_3_ and were treated as an exception, we obtained a final slope value equal to −0.69 (CI95%: −0.65 to −0.72) (Figure 1). Thus, the average antibody titer decrease from its highest peak (T_2_) six months post-vaccination is approximately 70%. 

After stratifying for age (Table 2), the youngest showed the smallest decrease with slopes equal to −0.54 (CI95%: −0.35 to −0.73), whereas HCPs aged 31 to 70 exhibited very similar linear regression slopes ranging from −0.66 to −0.74 (Table 2). The small group of HCPs older than 70 (*n* = 5) showed a larger, yet significant, antibody titer decrease equal to −0.88 associated to a rather large CI95% (−0.40 to −1.36) (Table 2). Stratifying for gender showed slopes equal to −0.66 (CI95%: −0.61 to −0.70) and −0.73 (CI95%: −0.68 to −0.79) for females and males, respectively (Table 2).

The 213 samples with titers above the 2500 U/mL instrument limit seemed to undergo the same 70% decrease as shown by the analysis of the 48 samples described later in Section 3.7 (Appendix A). Such samples, when needed, were diluted with pre-pandemic serum to obtain a titer within the instrument readout range. By applying the same criteria (i.e., excluding the HCPs showing a T_2_ to T_3_ antibody titer increase) we obtained an anti-RBD-S antibody titer decrease from T_2_ to T_3_ for the considered seronegative individuals equal to −0.64 ± 0.21 (Appendix A, Appendix A). It must be noted that 9 of the 48 HCPs were seropositive at T0 (Appendix A) and they showed an averaged T_2_ to T_3_ decrease equal to -0.85 (−0.81 to −0.89). 

### 3.7. Neutralization Assays and Diluted Samples

Forty-eight HCPs samples, nine of which were seropositive at T0 (Appendix A), were chosen at T_1_ to cover the anti-RBD titers instrumental range used in this study: 4 were below the 0.4 U/mL detection limit, 36 were within the instrument readout range and 8 were above the 2500 U/mL instrument limit (Appendix A). The 48 HCPs were also followed at T_2_ and T_3_. Each sample exceeding the 2500 U/mL instrument limit was diluted to 1:50 with a pool of pre-pandemic sera to obtain results within the quantification range. At T_1_, the 9 seropositive HCPs showed the well described exceptional antibody anti-RBD titers’ increase (Figure 3) (median 12,480 U/mL, IQR: 17,225 U/mL) whereas the 39 seronegative HCPs showed a median value approximately 500-folds lower (22.5 U/mL, IQR 266.8 U/mL). At T_2_, the seropositive HCPs showed an anti-RBD titers median value equal to 21,175 U/mL (IQR: 50,525 U/mL) approximately 16-fold higher than the seronegative one (1392 U/mL, IQR: 3737 U/mL), whereas at T_3_ the seropositive/seronegative median anti-RBD titers ratio drops to approximately 5.0 (2130 (IQR: 5448) and 427 (IQR: 986) U/mL for the seropositive and seronegative HCPs, respectively).

We also evaluated the neutralizing activity at T_2_ and T_3_ of the 48 collected sera (diluted to 1:80 and 1:160) against three SARS-CoV-2 strains isolated in our laboratory and belonging to the variants D614G, B.1.1.7, and B.1.351 (hCoV-19/Italy/LOM-UniSR-1/2020; hCoV-19/Italy/LOM-UniSR7/2021, and hCoV-19/Italy/LOM-UniSR6/2021 all available on GISAID sequence database—https://www.gisaid.org/ (accessed on 3 November 2021)). Results reported in Appendix A show that all tested isolates have a sensitivity profile to serum neutralization proportional to their corresponding anti-RBD titers. 

We calculated the ROC curve for the three variants by using the 1:80 data of both T_2_ and T_3_ (Figure 4). The highest AUC was observed for the D614G variant followed by the B.1.1.7 and the B.1.351 (Figure 4).

## 4. Discussion

Our study reports the antibody kinetics 6 months post-vaccination in both seronegative and seropositive individuals who received two doses of the BNT_1_62b2 vaccine. Seropositive HCPs showed, as already described, exceptional increase in antibody titer upon receiving the first vaccine dose [9,10,11,12,14]. In contrast, seronegative subjects showed the production of limited amounts of anti-RBD-S at T_1_, which was boosted by the second dose (T_2_), as described in Figure 1. Upon receiving the second dose (T_2_), the anti-S-RBD antibody titer also increased in individuals previously infected by SARS-CoV-2, reaching values approximately 10-fold higher than seronegative individuals. Six months after receiving the first dose, the average antibody titer decrease (from the T_2_ peak) was approximately 70% (within individuals aged 30 to 70). Such a decrease is less pronounced in young individuals in their twenties (~55% decrease). Although females showed a less pronounced antibody titer decrease than males, no statistically significant differences were observed between genders. Individuals who experienced COVID-19 before vaccination also showed a T_2_ to T_3_ decrease which, on average, is higher than seronegative individuals (~85%). We might speculate that the reason for this large decrease is due to the exceptional vaccine response, observed in this HCPs’ category, which cannot be sustained for a long time by the organism. 

Despite the 70% decrease (Figure 2), the mean antibody titer at T_3_ of the HCPs was still one order of magnitude higher than that of seropositive individuals before vaccination (T0).

In contrast to the general decreasing trend, a small subset of HCPs showed an increased antibody titer from T_2_ to T_3_. Such behavior, except for a few cases with a diagnosed post-vaccination infection, was observed mainly in older individuals with a low vaccination response. Further information is needed to inquire whether this behavior is related to a physiological slow vaccination response occurring in elderly individuals [33,34], or rather to an ongoing pathology/medical treatment. Notably, 12 HCPs were infected by SARS-CoV-2 post vaccination (Appendix A). Eleven out of twelve received two vaccination doses, whereas one was infected between the two doses. None of them showed severe symptoms, thus confirming the vaccines’ efficacy previously described in clinical trials [3]. Closer investigation showed that 8 out of 12 post-vaccination infected HCPs had been in close contact with a non-vaccinated SARS-CoV-2 positive family member. This observation further stresses that post-vaccination infection can be possible, even in the presence of a high anti-S serum antibody titer. Furthermore, the possibility of an in-hospital outbreak was ruled out since the 12 HCPs perform, within the hospital, different tasks.

Neutralizing activity experiments showed that the vaccine elicited antibodies are also effective against the B.1.1.7 and B.1.351 variants, yet, to a slightly lesser extent. A previous work [25] showed that in vitro, anti-RBD-S titers around 1400 U/mL provide protection from infection, yet HCPs with even higher anti-RBD-S titers at T_2_ were SARS-CoV-2 infected also shortly after the second dose. These data highlight both the difficulty to find a reliable correlate of protection by assessing the serum neutralizing antibody titers only, and, that other immune mechanisms, such as long-lived memory B and T cells play major protective roles against SARS-CoV-2 infection [35].

Our study, performed on well-stratified cohorts of individuals receiving COVID-19 vaccinations, highlights the importance of vaccination and stresses the need for further studies investigating the contribution of memory immunity after vaccination on protection from severe COVID-19.

## Figures and Tables

**Figure 1 vaccines-09-01357-f001:**
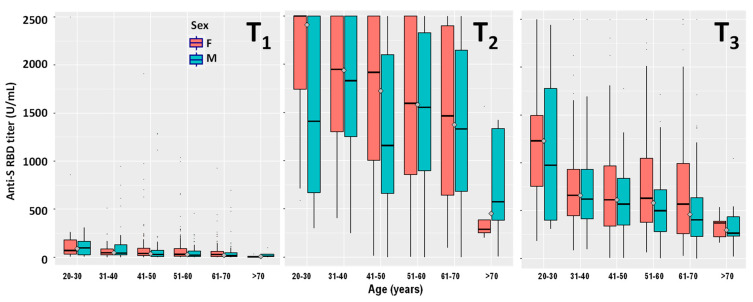
Stratification by age and gender of the serological responses at 21 days (**T_1_**), 42 days (**T_2_**), and 180 days (**T_3_**) after the first dose of the Comirnaty mRNA vaccine in HCPs never infected by SARS-CoV-2 (*n* = 958).

**Figure 2 vaccines-09-01357-f002:**
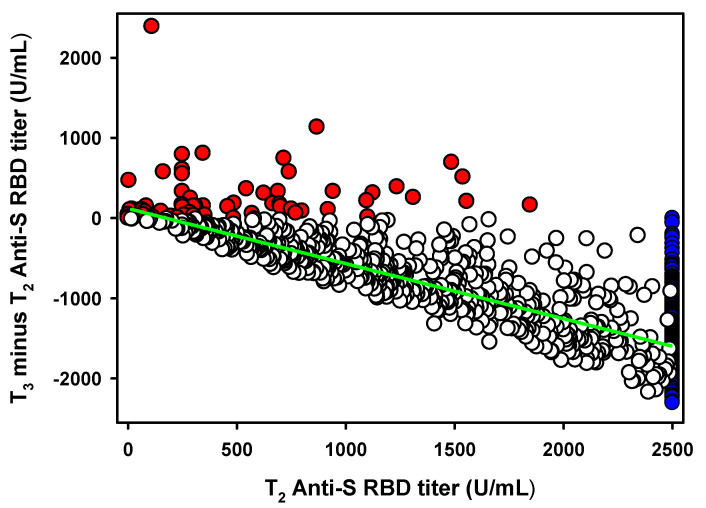
Association between antibody titer peak at T_2_ versus T_3_ minus T_2_ variation in the seronegative groups (*n* = 958). Red dots (*n* = 62) represent the small HCPs with an antibody titer increase between T_2_ and T_3_. Blue dots (*n* = 213) represent the HCPs showing an antibody titer >2500 U/mL at T_2_ or T_3_. Empty dots (683) represent HCPs with an antibody titer decrease between T_2_ and T_3_. The green line represents the linear regression analysis.

**Figure 3 vaccines-09-01357-f003:**
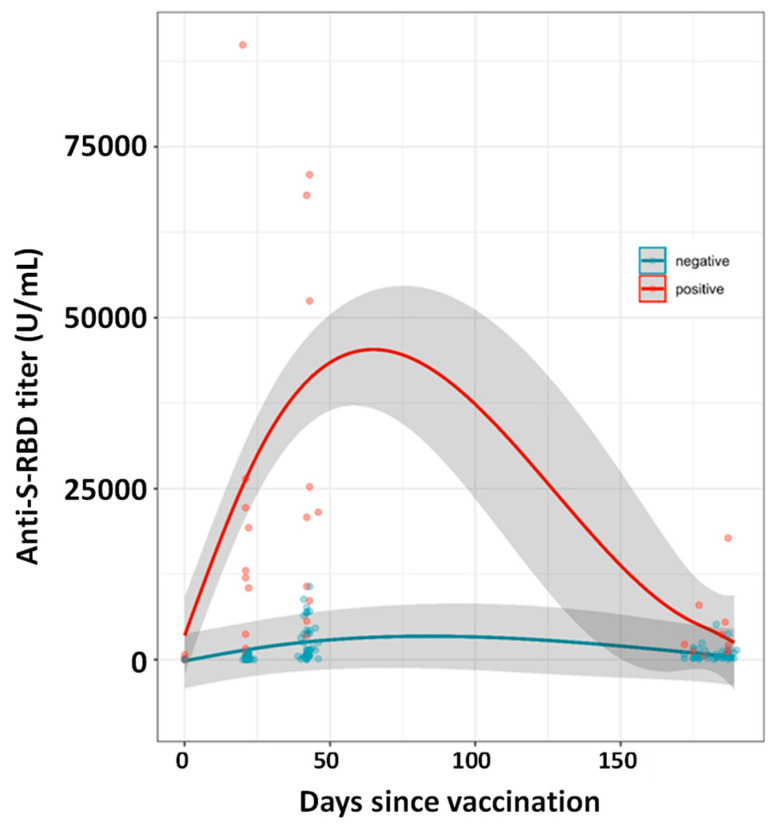
Kinetics of SARS-CoV-2 spike antibodies (U/mL) in seronegative (blue) and seropositive (red) individuals based on 48 HCPs (9 seropositives) described in Section 3.7. Samples above the 2500 U/mL instrument limit were diluted with pre-pandemic serum to bring the signal within the instrument readout range. The kinetics models are based on Generalized Additive Mixed effect Models with spline function as nonlinear term. Shadow area represents standard deviation.

**Figure 4 vaccines-09-01357-f004:**
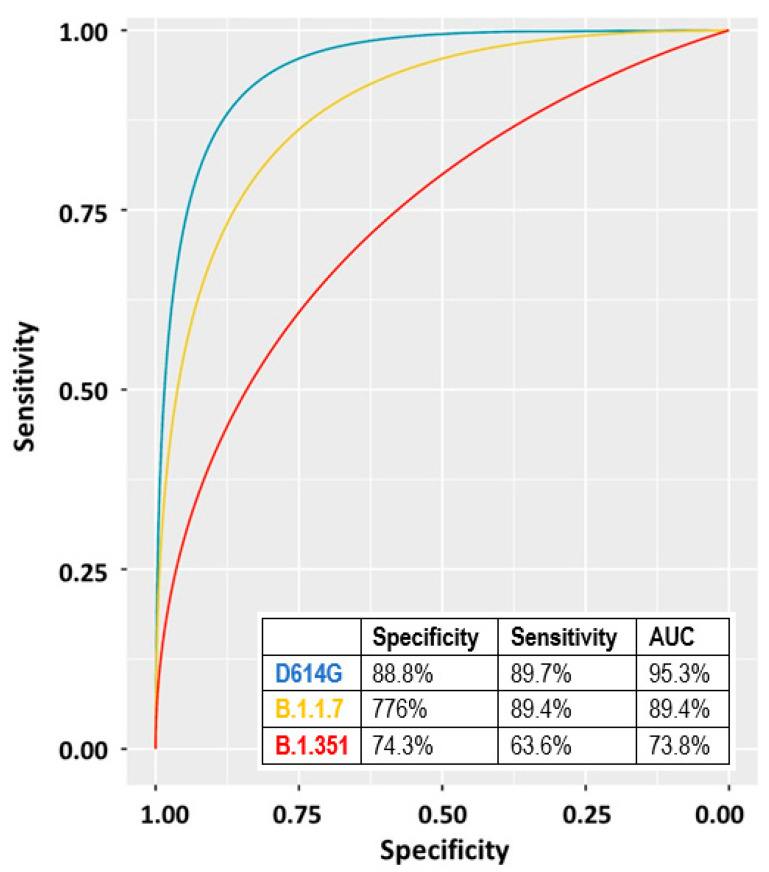
ROC curves and corresponding parameters obtained by considering the antibody titers of the 48 individual samples described in Section 3.7. Blue line: D614G, yellow line: B.1.1.7, red line: B.1.351.

**Table 1 vaccines-09-01357-t001:** Serological evaluation at 0 (T_0_), 21 (T_1_), 42 (T_2_) and 180 (T_3_) days post first vaccination dose. Test results are expressed as median (IQR). Results above the 2500 U/mL instrument limit were rounded to 2500 U/mL.

Subjects (Year ± SD)	N° ^a^	Test Results U/mL			
		T_0_	T_1_	T_2_	T_3_
**COV+** (44.3 ± 12.6)	82	58.3 (135.9)	2500 ^b^	2500 ^c^	2500 ^d^
**M** (44.2 ± 14.6)	30	42.9 (79.4)	2500 ^b^	2500 ^c^	2500 ^d^
**F** (44.4 ± 11.5)	52	67.2 (151)	2500 ^b^	2500 ^c^	2500 ^d^
**COV**− (49.4 ± 11.1)	958	/	31.9 (73.0)	1622 (1613)	591 (612.8)
**M** (51.1 ± 11.7)	319	/	27 (61.3)	1427 (1541)	499 (497)
**M** **20–30**	16	/	95.9 (141.3)	1408 (1831)	975 (1376)
**M** **31–40**	49	/	42.6 (104.7)	1829 (1249)	621 (518)
**M** **41–50**	61	/	28.3 (63.9)	1157 (1440)	568 (491)
**M** **51–60**	103	/	20 (57.3)	1554 (1432)	500 (436)
**M** **61–70**	83	/	16.5 (41.1)	1329 (1466)	404 (403)
**M** **71–80**	7	/	9.6 (99.8)	573 (949)	265 (201)
**F** (48.5 ± 10.7)	639	/	35.8 (75.3)	1755 (1569)	645 (658)
**F** **20–30**	39	/	69.4 (147.8)	2500 ^e^	1229 (741)
**F 31–40**	98	/	45.5 (58.6)	1947 (1199)	660 (483)
**F 41–50**	172	/	38.2 (78.2)	1916 (1496)	614 (628)
**F 51–60**	235	/	29.6 (79.9)	1595.5 (1646)	630 (668)
**F 61–70**	90	/	26.7 (54.9)	1464 (1757)	568 (734)
**F 71–80**	5	/	3.4 (4.2)	287 (136)	370 (164)

^a^ The 12 HCPs who got post-vaccine infection were omitted from this table. ^b^ Median was above the 2500 U/mL higher instrument limit. Of the 82 test results 11 (2 males and 9 females) were below the limit. ^c^ Median was above the 2500 U/mL higher instrument limit. Of the 82 test results 4 (all females) were below the limit. ^d^ Median was above the 2500 U/mL higher instrument limit. Of the 82 test results 36 (10 males and 26 females) were below the limit. ^e^ Median was above the 2500 U/mL higher instrument limit. Of the 46 test results 22 were below the limit.

**Table 2 vaccines-09-01357-t002:** Linear regression parameters calculated by analyzing the antibody peak at T_2_ versus the subsequent antibody titers decrease (T_3_-T_2_ serological evaluation).

Subjects’ Age Interval	N ^a^	Linear Regression Parameters		
		Slope (95%CI)	*p* Value	R^2^
**20–30**	31	−0.54 (−0.35 to −0.73)	<0.0001	0.533
**31–40**	95	−0.74 (−0.63 to −0.85)	<0.0001	0.660
**41–50**	151	−0.72 (−0.66 to −0.79)	<0.0001	0.769
**51–60**	230	−0.69 (−0.64 to −0.75)	<0.0001	0.723
**61–70**	101	−0.66 (−0.59 to −0.73)	<0.0001	0.778
**71–80**	5	−0.88 (−0.40 to −1.36)	<0.05	0.912
**Males**	227	−0.73 (−0.68 to −0.79)	<0.0001	0.770
**Females**	386	−0.66 (−0.61 to −0.70)	<0.0001	0.692
**Total**	613	−0.69 (−0.65 to −0.72)	<0.0001	0.720

^a^ Analysis was limited to the 615 seronegative HCPs showing antibody titers within the instrument readout range and decreasing at T_3_.

## Data Availability

Data available on request due to restrictions eg privacy or ethical.

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
