# Peer review of "Antibody Titer Kinetics and SARS-CoV-2 Infections Six Months after Administration with the BNT162b2 Vaccine"

_vaccines, 2021, doi:10.3390/vaccines9111357_

Round 1

Reviewer 1 Report

Antibody titer kinetics and SARS-CoV-2 infections six months after administration with BNT162b2 vaccine.

In this manuscript, Davide Ferrari et al present the results of the COVIDIAGNOSTIX study that was evaluating antibody response in 1052 Health Care Professionals who received BNT162b2 vaccination. In addition to a titer of antibodies authors also examined the neutralization activity of the blood against 3 virus variants.

This manuscript is presenting a large and comprehensive study that is very important in a current situation with pandemic and discusses questions about the development and stability of the anti-viral humoral response after receiving BNT162b2 vaccine. This manuscript additionally contains some material about the response to the viral infection in the subgroup from the studied population.

The major comments:

  1. One of the major comments is on the presentation and discussion of the results. Overall there is a lot of information presented in the plain text that is not reader-friendly for understanding. For instance, section 2.2 “inclusion criteria”  probably will benefit from the addition of the schema or table illustrating what cohort of HCP was receiving 2 doses of vaccine vs one dose and how the exclusion was affecting the numbers in studies.
  2. The authors provided a very detailed description of results in section 3 “Results”, but it does not always give the best depth of the material, for instance, part 3.5 (Post-vaccination Infection) has no table or graphics to refer to and it is making it harder to conceptualize presented information.   This part will benefit from some schema or table with the outcomes for the infected cohort.
  3. In contrast to “results” the discussion section would benefit from more discussion of the described results and also need to have references to the tables and graphs as illustrations of the points. Very confusing also is the presentation of the results for the booster with the second vaccination that is not described in the testing regiment.  The absence of a clear illustration makes it hard to understand the situation with the infected cohort.
  4. Figure 4 is very hard to read and the results section has very few explanations about it. It would be better to describe it in greater detail or omit it completely if it is not critical for this discussion.

Overall this manuscript has a large and very interesting set of results but will benefit from in-depth review and reorganization of the material and its presentation.

Minor suggestions:

  1. In the methods section 2.2 – it is not very clear what population received the second dose of vaccine and
  2. The methods section (lines 130-131) would be good to add more details about the format for the infection assay and CPE test.
  3. It would be good to have a second read-through of the manuscript for minor grammar corrections and improvement so the style. (for instance line 379)

Author Response

Reviewers’ comments

We thanks the reviewers for their valuable suggestions, which have been incorporated in the manuscript. We found the reviewers’ comments very pertinent and we believe that they greatly improved the quality of the manuscript.

A point-by-point reply to the reviewers' comments can be found below (in red).

Changes within the manuscript have been highlighted in red to help the reviewers’ work.

Reviewer 1

In this manuscript, Davide Ferrari et al present the results of the COVIDIAGNOSTIX study that was evaluating antibody response in 1052 Health Care Professionals who received BNT162b2 vaccination. In addition to a titer of antibodies authors also examined the neutralization activity of the blood against 3 virus variants.

This manuscript is presenting a large and comprehensive study that is very important in a current situation with pandemic and discusses questions about the development and stability of the anti-viral humoral response after receiving BNT162b2 vaccine. This manuscript additionally contains some material about the response to the viral infection in the subgroup from the studied population.

The major comments:

  1. One of the major comments is on the presentation and discussion of the results. Overall there is a lot of information presented in the plain text that is not reader-friendly for understanding. For instance, section 2.2 “inclusion criteria”  probably will benefit from the addition of the schema or table illustrating what cohort of HCP was receiving 2 doses of vaccine vs one dose and how the exclusion was affecting the numbers in studies.

We apologize if the inclusion criteria were not clear. All of the HCP received 2 doses of the vaccine (none received only one dose), and no exclusion criteria were applied. We modified the phrase in section 2.2 to better clarify such information. For the above reason (all of them received two doses) a table or schema was not necessary.

  1. The authors provided a very detailed description of results in section 3 “Results”, but it does not always give the best depth of the material, for instance, part 3.5 (Post-vaccination Infection) has no table or graphics to refer to and it is making it harder to conceptualize presented information.   This part will benefit from some schema or table with the outcomes for the infected cohort.

Thanks for the suggestion. A Table was added in the supplementary material and the manuscript text was changed accordingly.

  1. In contrast to “results” the discussion section would benefit from more discussion of the described results and also need to have references to the tables and graphs as illustrations of the points. Very confusing also is the presentation of the results for the booster with the second vaccination that is not described in the testing regiment.  The absence of a clear illustration makes it hard to understand the situation with the infected cohort.

Thanks for the comment. The boosting effect is simply the effect elicited by the second dose and observed at T2. We understand the word “boosting” might have created confusion and was omitted. We thus changed the title of the 3.6 section and we tried to better explain it in the results and discussion section. The figure describing the effect of the second dose is figure 1 where you can clearly see the increase from T1 to T2. This was better explained in the text. I hope this fulfilled your request.

  1. Figure 4 is very hard to read and the results section has very few explanations about it. It would be better to describe it in greater detail or omit it completely if it is not critical for this discussion.

Overall this manuscript has a large and very interesting set of results but will benefit from in-depth review and reorganization of the material and its presentation.

Figure 4 was moved to the supplementary materials.

Minor suggestions:

  1. In the methods section 2.2 – it is not very clear what population received the second dose of vaccine

All of the HCP received the second dose. We added a phrase to better explain this data

  1. The methods section (lines 130-131) would be good to add more details about the format for the infection assay and CPE test.

Thanks for the suggestion. More details were added in section 2.5.

  1. It would be good to have a second read-through of the manuscript for minor grammar corrections and improvement so the style. (for instance line 379).

Thanks for the suggestion. Line 379 was modified and a second-read through of the manuscript was performed.

Reviewer 2 Report

The article “Antibody titer kinetics and SARS-CoV-2 infections six months 2 after administration with BNT162b2 vaccine” written by Ferrari et al., shows SARS-CoV2 specific Ab levels at different time points in healthcare professionals after receiving the mRNA vaccine. Their methods, results and interpretations are clear but certainly need to address few questions and rewrite texts before acceptance.

  1. Introduction – “A few studies also evaluated the response after the second vaccine administration, however, the follow-up of enrolled subjects was up to 3 months only”. This is not correct as in June 2021, Doria-Rose published (https://pubmed.ncbi.nlm.nih.gov/33822494/) where 6 months follow-up was carried after mRNA vaccination (Moderna) and a different study where 6months after Pfizer mRNA vaccination was reported https://www.nejm.org/doi/full/10.1056/NEJMoa2110345. Include these references and rewrite the study cohorts are different.
  2. Also, the author didn’t describe enough about the study population being only healthcare professionals. Any specific rationale behind it or is the sample availability for a different study? Also, any specific reason for 6 months not later or early time point after the vaccine administration?
  3. Methods 2.2 = previous diagnostic tests were used to discriminate between SARS-CoV-2 previously infected and non-previously infected individuals. Kindly mention whether it is RT-PCR or any in-house methods?
  4. Ab levels are high in females at T0 in seropositive group, no other time point showed such gender related high levels. Did author see any such trend in post-vaccination infection group? It is not mentioned in the text.
  5. Result 3.6 = 958 seronegative groups’…represented by T2 (16). 16 in parenthesis is confusing, is it days after the vaccination, if so then it should be 42, not 16?
  6. Author didn’t write/discuss about decreasing in titer in related to the gender. Males seem to decrease compared to females. Also, I missed whether any decrease in titers among the seropositive groups were observed? I expedite the decrease would be slower compared to vaccinees, but kindly provide the details.
  7. I agree to “Neutralizing activity experiments showed that the vaccine elicited antibodies are also effective against the B.1.1.7 and B.1.351 variants” but increased activity against Alpha is something I would reevaluate as their Fig 4 and ROC curve (Fig 5) still shows Wuhan being high and alpha scoring next. Am I missing something here? Did author tested more sensitive live-virus neutralization for these selected 48 samples before making such interpretations? Table 3, the value should be 77.6% not 776% for alpha
  8. Overall, the data set is huge and it helps to display straightforward rather doing additional calculations/modified displays, kindly explain the advantage of displaying T2 vs T3-T2, are you considering T2 as control and T3 as test and control-test is reported?

Author Response

Reviewers’ comments

We thanks the reviewers for their valuable suggestions, which have been incorporated in the manuscript. We found the reviewers’ comments very pertinent and we believe that they greatly improved the quality of the manuscript.

A point-by-point reply to the reviewers' comments can be found below (in red).

Changes within the manuscript have been highlighted in red to help the reviewers’ work.

Reviewer 2

The article “Antibody titer kinetics and SARS-CoV-2 infections six months 2 after administration with BNT162b2 vaccine” written by Ferrari et al., shows SARS-CoV2 specific Ab levels at different time points in healthcare professionals after receiving the mRNA vaccine. Their methods, results and interpretations are clear but certainly need to address few questions and rewrite texts before acceptance.

  1. Introduction – “A few studies also evaluated the response after the second vaccine administration, however, the follow-up of enrolled subjects was up to 3 months only”. This is not correct as in June 2021, Doria-Rose published (https://pubmed.ncbi.nlm.nih.gov/33822494/) where 6 months follow-up was carried after mRNA vaccination (Moderna) and a different study where 6months after Pfizer mRNA vaccination was reported https://www.nejm.org/doi/full/10.1056/NEJMoa2110345. Include these references and rewrite the study cohorts are different.

We add the first papers in the introduction section (although was about a different vaccine: Moderna) whereas the second one, although using the Pfizer vaccine, does not monitor the serologic response but rather the efficacy based on the number of subjects who got infected post-vaccination.

This is consistent with what we wrote earlier that, to the best of our knowledge, no study describing the antibody titer kinetics six month after vaccination with the Pfizer vaccine were available.

  1. Also, the author didn’t describe enough about the study population being only healthcare professionals. Any specific rationale behind it or is the sample availability for a different study? Also, any specific reason for 6 months not later or early time point after the vaccine administration?

HCP were the first category who received the vaccine. Furthermore, being employees of the same hospital where the study was performed, it was easier to reach them and reach a high adherence to the study. We chose 0, 21, 42, 180 (and hopefully 360, the third dose is making things a little more complicated) days as a compromise between having a reasonable amount of data for a good kinetic description and economical (and organizational) reasons. Our cohort is quite large with respect of other studies. This implies a large economical effort as well as a large organizational effort (HCPs enrolling and the need for personnel performing the blood withdrawal)  that prevented us on doing a larger number of data-points. 

  1. Methods 2.2 = previous diagnostic tests were used to discriminate between SARS-CoV-2 previously infected and non-previously infected individuals. Kindly mention whether it is RT-PCR or any in-house methods?

Thanks for the suggestion. Yes, we used RT-PCR tests performed routinely within the hospital (added to the text).

  1. Ab levels are high in females at T0 in seropositive group, no other time point showed such gender related high levels. Did author see any such trend in post-vaccination infection group? It is not mentioned in the text.

Antibody titers at T0 depends strongly on when the individuals experienced COVID-19. In other words an individual might have been infected one month before vaccination (thus showing, likely, a high titer at T0) whereas another individual might have experienced COVID-19 6-10 months before vaccination (thus showing, likely, a much lower titer at T0). For this reason a comparison at T0 cannot be performed unless data about the timing of the disease were available. Such information could be available for the majority of the HCPs (except for asymptomatic individuals), yet, such evaluation was out of the scope of our study which focused instead on the vaccination response.

  1. Result 3.6 = 958 seronegative groups’…represented by T2 (16). 16 in parenthesis is confusing, is it days after the vaccination, if so then it should be 42, not 16?

(16) does not refer to “days after vaccination” but is the literature reference number of the paper describing that “21 days after the second dose” represents approximately the antibody titers peak. We modified the phrase to avoid confusion.

  1. Author didn’t write/discuss about decreasing in titer in related to the gender. Males seem to decrease compared to females. Also, I missed whether any decrease in titers among the seropositive groups were observed? I expedite the decrease would be slower compared to vaccinees, but kindly provide the details.

Thanks for the suggestion. A description of the gender related antibody titer decrease and the decrease observed in seropositive individuals was added in the first paragraph of the discussion. A phrase about the antibody titer decrease observed in seropositive HCPs was also added in the result section

  1. I agree to “Neutralizing activity experiments showed that the vaccine elicited antibodies are also effective against the B.1.1.7 and B.1.351 variants” but increased activity against Alpha is something I would reevaluate as their Fig 4 and ROC curve (Fig 5) still shows Wuhan being high and alpha scoring next. Am I missing something here? Did author tested more sensitive live-virus neutralization for these selected 48 samples before making such interpretations? Table 3, the value should be 77.6% not 776% for alpha

Thanks for the comment. The discrepancy between the AUC and the neutralizing thresholds is due to the different statistical mechanism underlying the two data. In simple words, the thresholds depends mainly on the first part of the ROC curve. If we magnified the initial part of the ROC curve we would see that the B.1.1.7 is steeper than the D614G (which leads to a smaller threshold). Yet, if we consider the whole graph, the AUC of the B.1.1.7 is smaller than the D614G. Although the 48 sample are quite an effort when performing neutralizing activity experiments, we agree that, statistically meaning, it might still not be a significant number. For this reason (and also because both reviewers said that the manuscript was a little crowded of information) we preferred to eliminate the part describing the calculated thresholds. We hope this will make the manuscript easier to read.

Table 3 was eliminated and its data were included in Figure 4 (77.6% was corrected accordingly).

  1. Overall, the data set is huge and it helps to display straightforward rather doing additional calculations/modified displays, kindly explain the advantage of displaying T2 vs T3-T2, are you considering T2 as control and T3 as test and control-test is reported?

T2 is considered (approximately) the antibody titer peak after 2 vaccination doses (see reference (16). Thus, T3-T2 is the antibody titer decrease from its maximum. Displaying T2 vs T3-T2 shows whether a statistically significant association exists between the antibody titer peak and its time-related decrease (we found that there is a strong association consistent with an approximately 70% decrease).